# Modelling Impacts of Nature-Based Solutions on Surface Water Quality: A Rapid Review

Fábio André Matos [1,*] and Peter Roebeling [1,2]

1 Centre for Environmental and Maritime Studies (CESAM) & Department of Environment and Planning, University of Aveiro, 3810-193 Aveiro, Portugal; peter.roebeling@ua.pt
2 Wageningen Economic Research, Wageningen University and Research (WUR), 6708 PB Wageningen, The Netherlands
* Correspondence: fabiomatos@ua.pt; Tel.: +351-963-631-613

**Abstract:** Global climate change and growing urbanization pose a threat to both natural and urban ecosystems. In these, one of the most impacted elements is water, which is responsible for a large variety of ecosystem services and benefits to society. Mathematical models can be used to simulate the implementation of Nature-Based Solutions (NBSs), thus helping to quantify these issues in a practical and efficient manner. This paper presents a rapid review of literature in which the effects of NBS on water quality were assessed with the help of modelling methods. It was found that only 14 papers deal with the topic in regard to NBSs. Most of these papers were published in European countries, using Nitrogen and/or Phosphorus as the studied water quality indicators and focusing predominantly on wetlands. The literature suggests that NBS can positively impact surface water quality, even under future climate conditions, while being a justified investment from an economic standpoint. It is suggested that more information is required in order to expand the evidence base on the effectiveness of NBS for water quality improvement as well as to develop better and more standardized methods to model NBS impacts on water quality.

**Keywords:** water quality; nutrient; sediment; Nature-Based Solutions; hydrological model

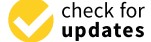



## 1. Introduction

It is expected that up to 68% of the world's population will be living in cities by the year 2050 [1]. This growing urbanization results in increased impermeable surface area, which leads to increased runoff, peak flow, and pollutant loads and concentrations [2,3]. The population is increasingly exposed to toxins resulting from industrial activities as well as traffic [4]. Urban non-point pollution is rising in urban environments, increasing concentrations of nitrogen, phosphorus, heavy metals, and organic carbon, among other pollutants, and is now a major case of concern for ecosystems in developed landscapes as well as human health [5–7].

With the growing recognition that hydrological resources must be carefully managed, water-related issues and potential ways to mitigate them have been a focus of study across several scientific areas, such as Environmental Science, Civil Engineering, and Hydrology. For this reason, the terminology used for solutions to urban water problems varies significantly in the literature, with terms employed such as 'Blue-Green Infrastructure' (BGI), 'Stormwater Control Measures' (SCM), 'Sustainable Urban Drainage Systems' (SUDS), 'Water Sensitive Urban Design' (WSUD), and 'Low Impact Development (LID) measures' often being used to refer to similar, or sometimes the same, applications. Despite the contextual differences in definition, these terms often overlap with the concept of 'Nature-Based Solutions'.

Nature-Based Solutions (NBSs) come in the form of natural or partly engineered measures that borrow from or incorporate nature in their usage as a means to mitigate societal issues. The International Union for Conservation of Nature (IUCN) defines NBS as

"Actions to protect, sustainably manage and restore natural or modified ecosystems that address societal challenges effectively and adaptively, simultaneously providing human well-being and biodiversity benefits" [8]. On the other hand, NBSs are defined by the European Commission [9] as "solutions inspired and supported by nature, which are cost-effective, simultaneously provide environmental, social and economic benefits and help build resilience". The natural focus of the definition, as well as the added social and economic aspects of the measures, separate NBSs from SUDS or LID. Nevertheless, the concept is considered open and not well-defined by the scientific community [10].

NBSs have been shown to be useful in the improvement of water quality [11]; their use in urban water management improves the quality of rainwater and runoff waters [12], and also helps create an approximation of the natural water cycle [13].

Despite the rise in NBS-related research observed in recent years [14], these measures are still slowly and scarcely implemented when compared to the more conventional grey infrastructure [15,16]. As NBS initiatives are becoming more common over time, there is a rising need to quantify and evaluate their impacts, which is reflected in an increased interest in hydrological modelling [17]. Modelling tools are used by design engineers to simulate flow conditions to better understand the performance of NBS and their configuration as treatment units [18]. Additionally, some of these come pre-loaded with the necessary tools and systems to model NBSs, something that is fairly recent as the software suites did not initially support such functions. Nevertheless, gaps in our understanding of ecohydrological processes, as well as the strict functionality of existing models, stand as limiting factors when it comes to NBS modelling [19].

This paper will perform a rapid review of literature where impacts of Nature-Based Solutions on surface water quality are modelled using any tool capable of predicting these phenomena with a significant level of accuracy.

## 2. Review Methods

### 2.1. Methodology and Search Parameters

The methodology employed in this paper was that of a rapid review carried out in a systematic manner and in accordance with the Preferred Reporting Items for Systematic Reviews and Meta-Analyses (PRISMA) guidelines [20]. In this way, the process is clearly detailed and documented in such a way that it can be replicated in order to obtain the same results [21]. Rapid reviews are increasingly used as a faster alternative to traditional systematic reviews, especially under time or financial limitations [22]. These reviews can be carried out in a fraction of the time required to perform their systematic counterparts without significant impact on their conclusions, at the cost of limiting the scope of the search [23]. In this case, the limitations comprise the use of only peer-reviewed articles indexed in one electronic database.

The literature search was performed on 11 November 2021 using the Scopus scientific database (https://www.scopus.com/ accessed on 11 November 2021). The focus of the search was on scientific papers written in English and directly related to the modelling or simulation of water quality parameters in the context of Nature-Based Solutions (NBSs). Additionally, only papers published in scientific journals were reviewed, with any grey literature not being considered when making this review. The temporal window of publication was not set; however, because NBSs are a new topic of research, this limitation was not expected to impact the search results.

The specific parameters used were a combination of terms that had to appear in the paper's title, abstract, or keywords. These terms were "Nature-Based Solution(s)" with and without hyphen (Nature-based solution), and any occurrence of the following terms, which are commonly present in water quality studies: water quality, water pollution, water pollutant(s), nutrient(s), nitrogen, nitrate*, NO3, phosphat*, suspended solid(s), TSS, sediment(s), or contaminant(s). Additionally, the results needed to include at least one of the following terms: model*, simulat*, or estimat*. The words with an asterisk represent terms based on a shorter string, such as "model" and "modelling". After conducting

the initial search, the results were filtered by source type in order to obtain only journal publications, and then by language to filter out any non-English literature (albeit none occurred). This process is illustrated in Figure 1.

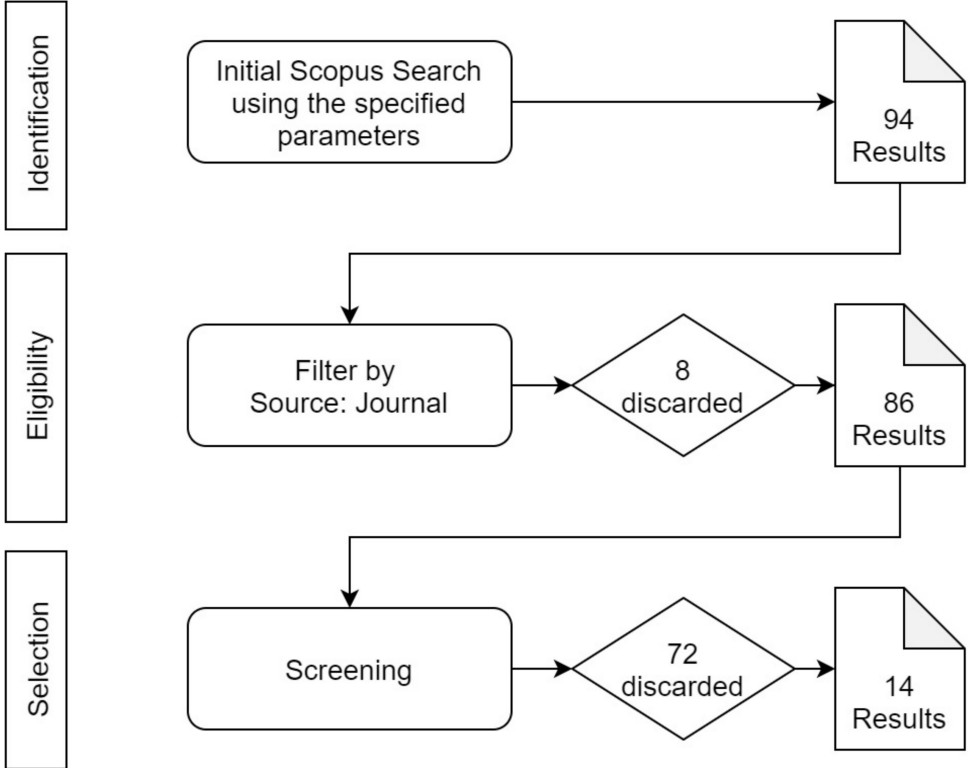

**Figure 1.** Diagram illustrating the literature search and selection process.

The list of publications resulting from the search was then perused in order to ascertain which papers were relevant for this study. The titles and abstracts were read and selected/excluded based on relevancy to the subject in question, with a more in-depth read of the paper sometimes being necessary when the details of the study were not clear from the abstract alone. Both authors took part in this selection process in order to avoid bias. All selected papers were then thoroughly analyzed for the purposes of writing this review.

Due to the relatively low number of publications to be selected for analysis, the authors decided to conduct further searches in order to verify if the terms used were the most appropriate for water-quality-centered studies. The hypothesis was that other commonly used terms, such as Sustainable Urban Drainage Systems (SUDS), Blue-Green Infrastructure (BGI), or Water-Sensitive Urban Design (WSUD), would yield more results than the term NBS. Therefore, three separate searches were conducted (20 November 2021) in which the primary search terms (NBS, Nature-Based Solutions) were changed to the new ones, while keeping all the other keywords as they were initially. The results were also immediately filtered by source, looking for Journal Publications only. Thus, results were generated for searches including "SUDS" and "Sustainable Urban Drainage Systems"; "GBI", "Green-Blue Infrastructure, "BGI", and "Blue–Green Infrastructure" (both with and without the hyphen); and "WSUD" and "Water Sensitive Urban Design", respectively (see Figure 2).

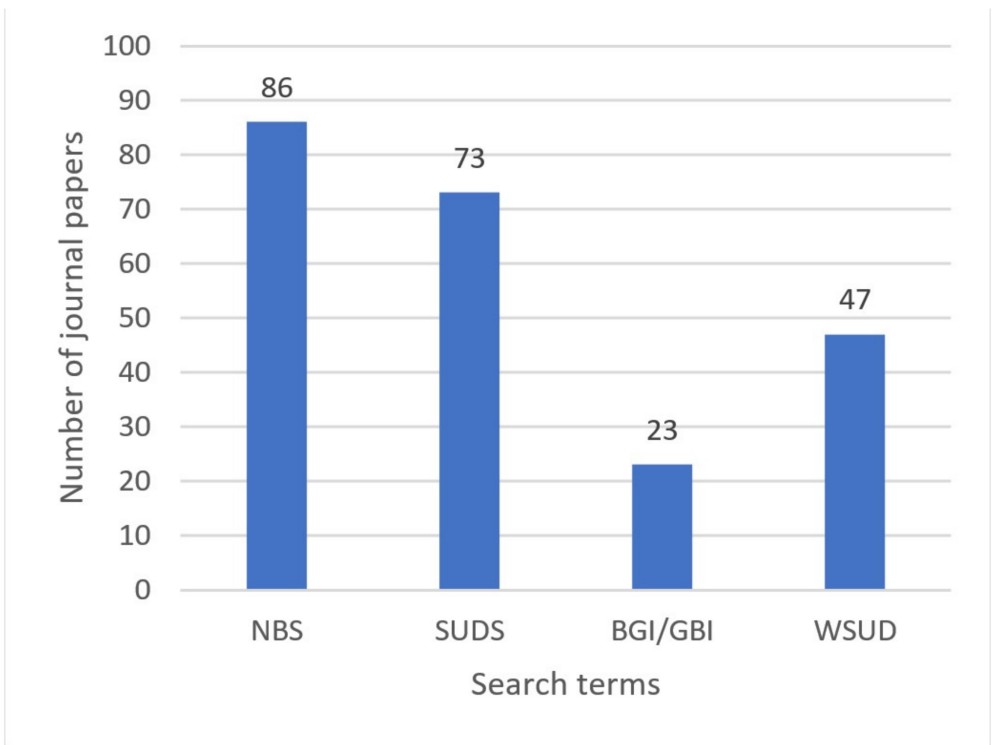

**Figure 2.** Journal publication search results for the most common terms analogous to Nature-Based Solutions (NBSs), such as sustainable urban drainage systems (SUDS), blue-green infrastructure (BGI)/green-blue infrastructure (GBI), and water sensitive urban design (WSUD).

*2.2. Definition of Nature-Based Solutions*

The scientific community still considers NBSs to be an open concept with a meaning that may appear vague with unclear links to pre-existing concepts [10,24], thus, needing further clarification [10]. For this reason, it is essential to define what is considered an NBS for the purposes of this review. Along with other similar measures, the concept of NBSs promotes the maintenance, enhancement, and restoration of ecosystems as a means to simultaneously address several societal challenges [25]. The core definitions from IUCN [8] and the European Commission [9] presented in the Introduction express the essential ideas behind the concept of NBSs, but each focuses on substantially different aspects of these actions. Sowińska-Świerkosz and García [26] reviewed several definitions of the term and defined NBSs as interventions that are inspired and powered by nature, address societal challenges or resolve problems, provide multiple services/benefits, including biodiversity gain, and are of high effectiveness and economic efficiency; arguing that measures should fulfil most, if not all, of these parameters to be considered an NBS.

The authors of this paper consider NBSs as any practical intervention or structure incorporating or revolving around natural elements that aims to address an environmental concern. In this study, those would mainly be related to water treatment or pollution load reduction. In this way, solutions such as constructed wetlands, retention basins, bioswales, or riparian buffers are all considered NBSs, while rain barrels and first-flush tanks are not. This simplification of the broad and complex definition of NBS is used here to make the most of the limited pool of articles resulting from the literature search. In this way, the review can incorporate studies simulating the implementation of measures that some authors may not consider NBS, such as permeable pavements.

*2.3. Definition of Modelling*

Modelling is, generically, the act of using mathematical means to calculate or predict the outcome of a complex system or process. The physical state of media (such as velocity, pressure, density, etc.) at every point in a specific domain can be predicted using numerical formulations, and models can solve the state of an entire system using sophisticated but well-understood numerical equation solvers [27]. The word 'model' is used to refer to a number of concepts, from theoretical frameworks to components of a process or piece of software.

Clear definitions of the general concept of modelling are hard to find; however, when analyzing modelling applications in specific areas of science, more concrete definitions can be found. Lin and Beck [28] state that "the structure of a model can be defined by the input, state, and output variables chosen to characterize the behavior of the modeled system". The definition that better fits this study is that of hydrological modelling, which is described by Allaby and Allaby [29] as the characterization of real hydrological features and processes using small-scale physical models, mathematical analogues, or computer simulations. Thus, this type of modelling is used in the study of environmental phenomena concerning water excess or scarcity, as well as dissolved or solid material transport [30]. This definition encompasses both the use of simple mathematics and custom models in one dimension to the more complex processes with established software and applications, such as the Storm Water Management Model (SWMM; [31]) or the Soil & Water Assessment Tool (SWAT; [32,33]). As the topic of this study is water quality, the modelling being considered is, specifically, the quantitative estimation of numerical values of water quality indicators (such as pollutant loads and concentrations) for cases involving NBSs.

## 3. Results and Discussion

The initial literature search returned 94 papers, which were reduced to 86 after applying the source type and language filters. It should be noted that, despite applying the language filter, no papers were discarded as all papers were in English. An initial assessment was carried out by reading the titles and abstracts of each paper; then papers that seemed relevant were further analyzed with a full read in order to be categorized as eligible. In the end, 14 papers were considered relevant for this review (see Table A1 in Appendix A). It is evident from these results that there is a lack of papers on the subject of modelling Nature-based Solutions (NBS) impacts on water quality. Considering that similar studies are often carried out by Civil Engineering experts who prefer the usage of the terms "SUDS", "WSUD", "GI", or "BGI", this data scarcity can instead be interpreted as reflecting a scattering of available papers across the multiple terminologies, making an in-depth review hard to achieve. Conducting a Scopus literature search with the parameters "model", "water quality", and any occurrence of the aforementioned terms returned 1958 papers (on 15 October 2021), which helped to support this hypothesis.

The large number of discarded papers is attributed to the relatively broad scope of the initial search, as the word model can be used to describe various processes. This is confirmed by many discarded entries of this search, which focused on controlled experiments in laboratory-designed conditions or in real-world field studies, as well as papers mentioning economic models that are not the modelling approach this study is looking for. With the inclusion of common pollutants such as nitrogen, several soil and air pollution papers appeared that were also discarded, as were papers pertaining to soil erosion and deposition that did not model transported loads (which were likely resulting from the term "sediment" in the search parameters).

*3.1. Spatial Distribution*

This study found that most of the literature on the topic of modelling NBS impacts on water quality is published in European countries (Figure 3, left); over 55% of publications came from Europe and, of those, over 33% originated from Italy. Approximately 20% of the publications originated from the American continent, two-thirds of which came from the

United States of America and the remaining third from Brazil. Lastly came Oceania, with two publications from Australia, and Asia with one publication from China.

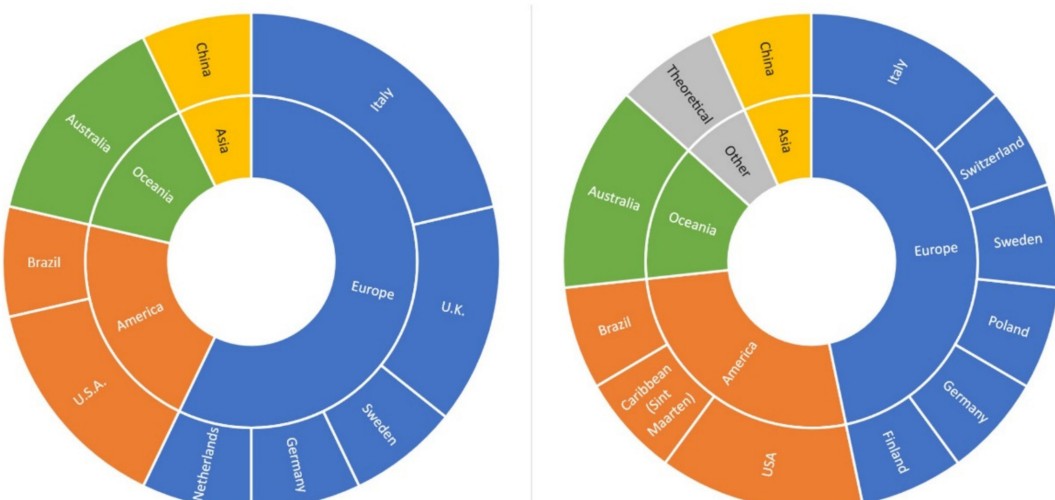

**Figure 3.** Visual representation of the global distribution of publications (**left**) and the distribution of case study locations across publications (**right**).

An important aspect to point out is that these results are highly biased towards European publications because of the usage of the term NBS, which is less common in other continents. This is also one of the reasons we see a relatively low number of papers published in the USA, which, similar to China, is one of the countries with the highest usage of water quality modelling methods in literature [34]. Indeed, the term NBS is one that is used predominantly in Europe, while other countries prefer the engineering terminologies (such as SUDS, WSUD, or BGI). For this reason, a large majority of European sources in the results of this literature search were expected and are not indicative that Europe has more studies on NBS-like water pollution management measures than other major contributors to these academic fields. Regardless, the numbers do show that Europe is concerned with environmental issues and that the intent to apply NBSs to the betterment of society is a priority.

*3.2. Case Study Location*

It was found that most authors (over 70%) wrote about case studies located in the country of publication (see Figure 3, right), with few authors writing about multiple locations. UK-published papers showed the most variability among the selected literature. Seyedashraf et al. [35] wrote about a theoretical case study not located anywhere in particular, and Zawadzka et al. [36] wrote about three cases in the same paper, of which only two were considered relevant for the purpose of this review (one in Poland and one in Switzerland). Another exception is the paper published in Italy by Gallotti et al. [37], which considered six case studies, two of them in Italy, and of which only the case study in Finland contained water quality modelling, thus being the only one considered in this review. Dutta et al. [18] published their paper in the Netherlands but speak of a case study in the Caribbean, i.e., Sint Maarten, which is considered a constituent country of the Kingdom of the Netherlands. Because the case studies are predominantly related to the countries of publication, they follow the same trends and, as such, the discussion regarding the European bias of the NBS terminology described in the previous section also applies here.

### 3.3. Temporal Distribution

The term NBS is a relatively recent one, with its usage going back no further than 2010, and only in 2015 was it officially defined by the European Commission [9]. As a result, the papers selected for this review are recent ones. The initial search results returned papers from 2016 to 2021, and after filtering for relevancy, the oldest papers in the selection go back no further than 2018 (see Figure 4). From there, we see a fast increase in total publications on the subject until the year 2021, while noting that not all publications from 2021 are included as this review was conducted in November 2021 (and still more papers are expected to have been published until December 2021). This recent interest in the subject is due to the popularity of the NBS topic as well as the, now more widespread, availability of open-source modelling software, such as the Integrated Valuation of Ecosystem Services and Tradeoffs (InVEST; [38]) or SWMM [31] models.

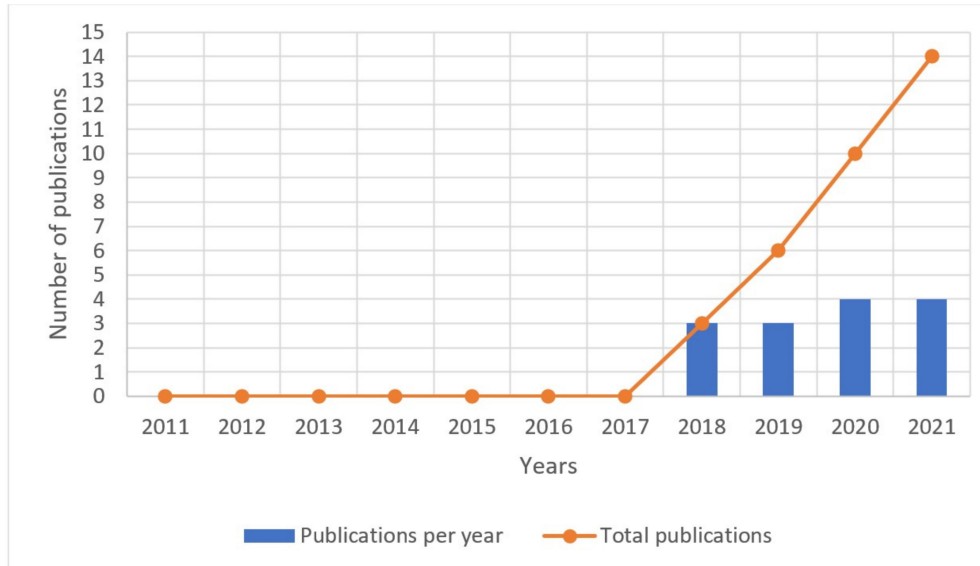

**Figure 4.** Graphical representation of the temporal distribution of publications. The orange line represents total publications while the blue line represents actual publications per year.

### 3.4. Spatial Scale of Analysis

In terms of the spatial scale of the case studies being analyzed, the papers differ from one another, although there is a clear predominance of large-scale studies (see Figure 5).

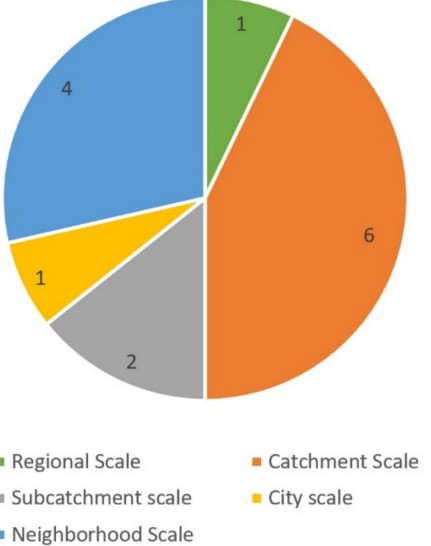

**Figure 5.** Visual comparison of the number of publications dealing with spatial scale of case studies.

Approximately 60% of the publications' case studies deal with large scales, be they regional, catchment, or sub-catchment scale; the remaining 40% focus on smaller areas such as specific neighborhoods or a city. Baustian et al. [39] studied the largest area, consisting of the entire Breton Sound Estuary (Lousiana, USA). Di Grazia et al. [40], Dutta et al. [18], Zhang et al. [41], Singh et al. [42], Fu et al. [43], and Castonguay et al. [44] all analyzed water quality at a catchment scale for a specific area, while Gallotti et al. [37] and Castañer et al. [45] studied specific sub-catchments. On the small-scale analysis side, Zawadzka et al. [36] studied several areas that are on the larger spectrum but are limited to the city scale for the water quality modelling. Case studies by Hamann et al. [46] and Masseroni et al. [47] focused on neighborhoods or similarly contained urban areas, Seyedashraf et al.'s [35] theoretical case study consisted of a synthetic small-scale urban drainage system, and Symmank et al.'s [48] case study consisted of a small section of a German river. The catchment-scale approach of the majority of these papers is expected, as the modelling of water quality parameters is usually associated with hydrologic models that are predominantly performed at these scales for best results. Nevertheless, larger-scale models are not always suitable to assess small-scale NBS implementation. Despite being important to capture the behavior of these solutions at small scales, studies focusing on neighborhood or local modelling of NBS effects on water quality are scarce. City-scale modelling studies are even rarer, and with a scope sitting between larger scales and the limited neighborhood ones, this scale holds great potential in determining how NBS may benefit urban environments and should be further explored.

### 3.5. NBS Assessed

Across the 14 papers analyzed in this review, 10 different NBS types are mentioned for a total of 28 solutions analyzed (see Figure 6), i.e., an average of 2 per publication.

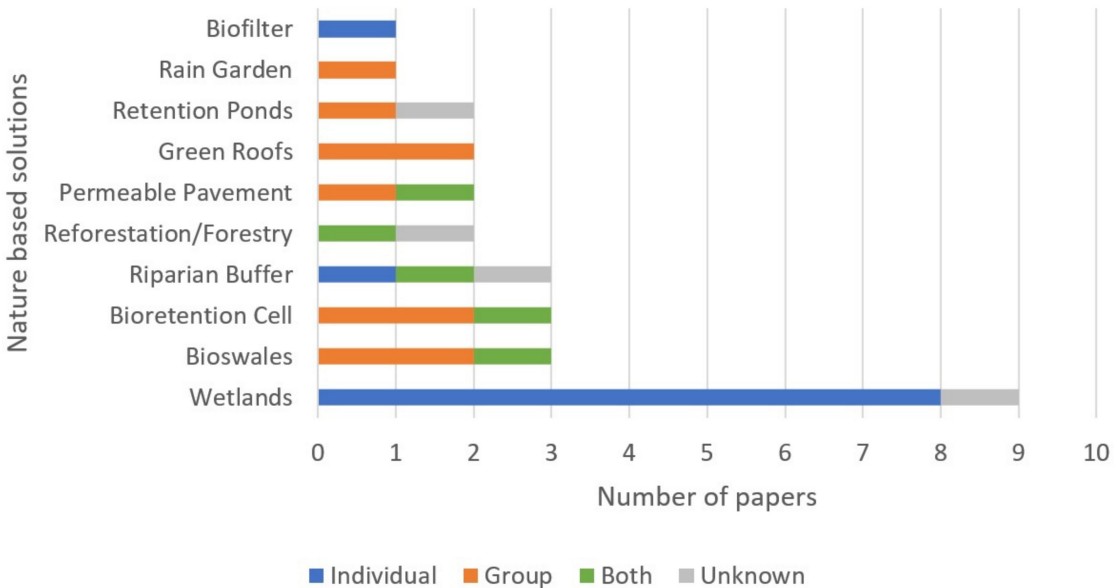

**Figure 6.** Graphical representation of the frequency of occurrence of each NBS in the literature reviewed. Some NBS impacts were assessed individually, others were sets of multiple NBS, and sometimes both approaches are taken.

Wetlands are by far the most frequently occurring NBS in water quality modelling studies, likely due to their extended use in wastewater treatment as well as stormwater control. Another interesting finding is that wetlands are most often the sole focus of the studies they appear in, as is the case in eight out of the nine papers mentioning wetlands in this review. Other solutions, however, appear often in groups, as is the case of green roofs (appearing in two papers) that are always accompanied by more solutions. This is also the case for permeable pavements, reforestation, bioretention cells, retention ponds,

and bioswales. The reason for this might be that these solutions (with the exception of bioretention cells) are more often used for air quality improvement and flood control, and less so for water treatment, which is the topic of this review. In this way, the focus of a paper may be on the modelling of NBS in sets or individually, but the focus may also be mixed. As an example, Dutta et al. [18] modelled the pollutant removal efficiency of bioretention cells, bioswales, and permeable pavements individually (in sets of two and sets of three), while Seyedashraf et al. [35] used model optimization to find the optimal scenario with multiple solutions, among which are permeable pavements, green roofs, rain gardens, bioswales, and other non-Nature-Based Solutions. It should be noted that Gallotti et al. [37] presented methodologies only, leaving results for a future study, without mentioning how each NBS will be assessed (NBSs occurring in this particular paper appear as the type 'unknown' in the aforementioned graph). It is clear from Figure 6 that there is not enough data on how to model the individual impacts of several NBSs on water quality, mainly those that are not specifically related to the subject but could still have noticeable effects, as is the case of green roofs, but also of directly water-related solutions such as retention ponds and bioretention cells.

### 3.6. Modelling Methods Used

With 14 different studies from different locations, we expected the use of varied models and methods to make the necessary calculations (see Figure 7). With the present state of technology and the available water quality models, with some having already built-in mechanisms to consider NBSs, it is not surprising that the majority of these publications resort to one or more software tools to aid in this effort.

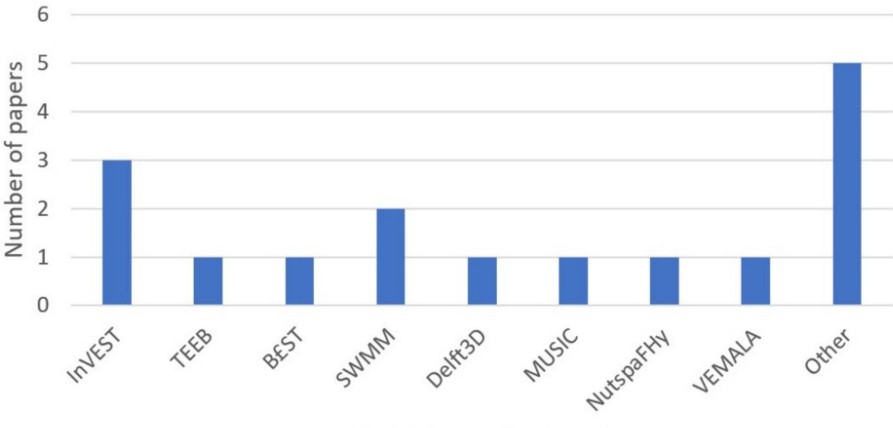

**Figure 7.** Comparison of frequency of use of each model encountered in the literature reviewed. Custom models or methods relying on manual calculation using mathematical equations fall in "Other". **Note:** InVEST refers to Integrated Valuation of Ecosystem Services and Tradeoffs [38]. TEEB refers to The Economics of Ecosystems and Biodiversity [49]. B£ST refers to Benefits Estimation Tool [50]. SWMM refers to Storm Water Management Model [31]. MUSIC refers to Model for Urban Stormwater Improvement Conceptualization [51].

The most commonly recurring modelling software used in these papers was Stanford University's and Natural Capital Project's 'Integrated Valuation of Ecosystem Services and Tradeoffs' (InVEST) suite [38], an ecosystem service valuation and mapping tool employed by Di Grazia et al. [40], Zwadazka et al. [36], and Singh et al. [42]. The InVEST tool comes with a nutrient delivery ratio model that uses a mass balance approach [52] to model nutrient dynamics based on source mapping, transport capacity, and retention values of different land use and land cover conditions of an area, as well as the morphology of the catchment. The second most popular modelling tool used was the United States Environmental Protection Agency (EPA)'s 'Storm Water Management Model' (SWMM), a software solution for the design and analysis of stormwater runoff control strategies [31].

SWMM is reportedly the tool most commonly used by researchers to model NBSs and water quality [18], a trend that is not verified in this review as InVEST is used in one more paper than SWMM. The tool comes with hydraulic modelling and pollutant load estimation capabilities, being employed by Dutta et al. [18] and Seyedashraf et al. [35]. Hamann et al. [46] chose to use two different modelling tools, the Dutch National Institute for Health and Environment's 'The Economics of Ecosystems and Biodiversity' (TEEB; [49]) and SusDrain's 'Benefits Estimation Tool' (B£ST; [50]). Both tools follow an ecosystem modelling approach as in InVEST, as opposed to the hydraulic modelling approach of SWMM. The author of the aforementioned paper modelled their water quality parameters using both tools, comparing the results after validation. Baustian et al. [39] used the Deltares Systems' Delft3D, the 3D modelling suite for hydrodynamic, sediment transport and water quality modelling, more precisely, the Integrated Biophysical Model [53]. Another similar approach was taken by Zhang et al. [41] with eWater's 'Model for Urban Stormwater Improvement Conceptualization' (MUSIC), which contains a wide range of functions to model stormwater runoff and contaminant removal resulting from treatment devices [51]. Lastly, Gallotti et al. [37] used NutSpaFHy, a tool that can model nutrient export from forest areas, in conjunction with Finnish Environment Institute's VEMALA, which calculates nutrient loads from the remaining land use and land cover categories [54,55]. Three of the aforementioned software suits (37.5%), SWMM, B£ST, and MUSIC, are advertised as directly supporting the modelling of NBS or analogue measures, suggesting that practitioners have access to varied options for these sorts of studies even before having to consider less-specific tools.

A total of five papers modelled NBS impacts on water quality without relying solely on software. Symmank et al. [48] used two proxy-based models to estimate nutrient retention rates of NBS according to an extensive literature search from European case studies with empirical data. Castañer et al. [45] applied the P-k-C* model [56] to evaluate wetland nutrient removal efficiency using mathematical equations only. Castonguay et al. [44] used the Simple Method [57] to evaluate the nutrient removal efficiency of each solution. Lastly, making use of Geographic Information Systems (GIS), Fu et al. [43] relied on a custom model based on nutrient balance equations that was then run with the assistance of ArcGIS, while Masseroni et al. [47] used a mathematical model based on first-order degradation kinetics and their relation to the system design of the solutions provided through the use of QGIS.

### 3.7. Pollution Indicators Studied

Several elements exist that can be used to ascertain the quality of surface waters. Some commonly used elements are Nitrogen (N) and Phosphorus (P), commonly grouped into the term Nutrient Pollution; other indicators include suspended solids and biochemical oxygen demand. The primary pollutant elements studied in the 14 selected papers are shown in Figure 8.

Nitrogen appears as the most commonly used element in the studies, which is to be expected as it is, along with Phosphorus, the most studied pollutant in the field of surface water quality. Indeed, Nitrogen levels in the form of Total Nitrogen (TN) are studied in over half of the selected papers, and although some mention Nitrates and Kjeldahl Nitrogen (TKN), they end up not being part of the modelled elements. Another element that is sometimes present in such studies is ammonia, but it is also not mentioned in these publications, although its effect should also be included in the TN value. As is expected, the second most commonly studied element is Phosphorus in the form of Total Phosphorus (TP), appearing in half of the publications. This element is often studied in addition to N, but it also appears on its own in two papers as it is a good measure of agricultural water pollution. Sediment is also studied in four papers in the form of Total Suspended Solids (TSS), which is an easily observable measure of water quality. Biochemical Oxygen Demand (BOD) is not an element itself, but rather a value that can be used to estimate the quality of water and can also hint at the behavior of other pollutants, such as TN and

TP. For this reason, BOD appears as the water quality indicator modelled in two papers, although no other associated indicators such as Dissolved Oxygen (DO) are present. Lastly, Hamann et al. [46] use only 'water quality' as a general indicator since their focus is on ecosystem service valuation and not so much on pollutant load modelling. Despite being simplified, this metric allows the reader to obtain a general understanding of the effect of NBS implementation.

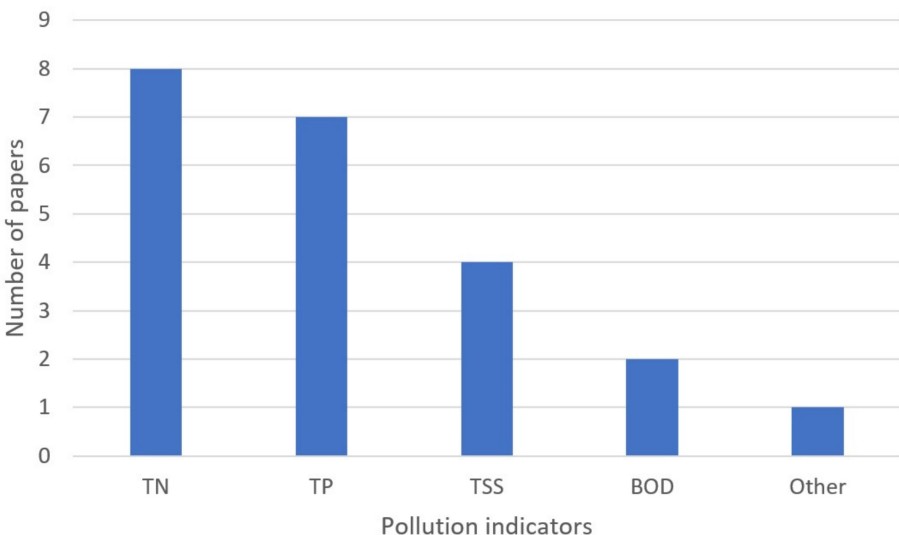

**Figure 8.** Graphical comparison of the frequency of occurrence of each main pollutant in the reviewed literature. **Note:** TN refers to total nitrogen. TP refers to total phosphates. TSS refers to total suspended solids. BOD refers to biochemical oxygen demand.

Results show that NBS can be beneficial to surface water quality by reducing the export of nutrient pollution (N, P) [36,40,42,43] as well as suspended solids [35,36], and by promoting nutrient removal services (N, P, BOD) [18,47,48]. Some authors observed that NBS had "excellent performance" [47], while others saw only small improvements to water quality [46], with no particular performance trends associated with specific elements. It is also important to note that results from these studies should not be generalized, as the models are sensitive and require that calculations consider the local circumstances of each case study [36,46].

### 3.8. Economic Aspects

Of the 14 papers studied in this review, 50% of them include some type of economic assessment in their respective studies. Di Grazia et al. [40] and Castañer et al. [45] performed cost–benefit analyses by assessing the monetary benefits of nutrient pollution reduction and removal as well as by providing estimates on the costs of NBS implementation in their respective scenarios. Similarly, Hamann et al. [46] focused on ecosystem service valuation using B£ST and TEEB, and estimated water quality benefits without focusing on particular pollution indicators, reaching monetary values for added benefits to water quality with the implementation of the studied solutions in relation to their costs.

Fu et al. [43] performed a benefit assessment in their estimation of the indirect monetary values associated with externalities from NBS preservation as opposed to their conversion for other purposes. The considered avoided damages are those for fisheries in the case study site.

The remaining three papers performed cost-effectiveness analyses (thus not considering the monetary benefits of NBS implementation). Seyedashraf et al. [35] made an assessment of the capital costs required to implement NBSs with varying degrees of performance in reducing TSS levels. Singh et al. [42] estimated the minimum capital costs required to restore wetlands in relation to their expected nutrient load reduction capacity.

Castonguay et al. [44] provided information regarding the implementation and maintenance costs of NBS under certain budget scenarios in their optimization process.

The most common types of economic assessments are related to the costs of implementation of NBSs, which is an important metric for stakeholders considering the implementation of NBS.

Overall, the literature reviewed suggests that the water quality improvements obtained from NBS implementation can provide substantial benefits. The cost–benefit analysis-based studies highlight the economic feasibility of these solutions [45], deeming them well-justified considering their long-term benefits relative to the investment necessary [40]. The benefit assessment study also acknowledges the economic value of externalities from the water quality improvements of NBSs [43]. Finally, the cost-effectiveness approach studies do not mention economic viability but estimate that significant water quality improvements can be achieved by NBSs, with larger investment values often being associated with better performances [35,42].

### 3.9. Future Climate Conditions

Only three out of the fourteen papers analyzed in this review considered future climate predictions in their studies. Di Grazia et al. [40] assessed water quality for baseline conditions and compared the results with those including NBS effects in combination with future precipitation and temperature conditions using the greenhouse gas Representative Concentration Pathway (RCP) 4.5 scenario. Gallotti et al. [37] and Zhang et al. [41] instead considered the higher-emissions scenario RCP8.5 in their calculations, using a similar approach considering both baseline and future climate condition scenarios with NBS effects.

While there is no significant pool of papers from which to extract conclusions on this topic, it is worth noting that Di Grazia et al. [40] found that NBSs can effectively reduce nutrient exports under RCP 4.5 conditions. Zhang et al. [41] found that NBS performed about the same under RCP 8.5 conditions as they do under current conditions, recommending larger systems in order to maintain water treatment effectiveness.

### 4. Conclusions

This paper provided a succinct analysis of the main publications tackling the topic of modelled effects of Nature-Based Solutions on surface water quality. The study found that the majority of publications originated from European countries (over 55% of publications), followed by the American continent, Oceania, and, lastly, Asia. The explanation for this European dominance lies in the specific search for the term 'Nature-Based Solutions' (NBSs), which is more commonly used in Europe than in other regions. It was also found that the majority of the case studies (over 70%) centered on the country of publication for the respective paper. The study found that publications on the topic go as far back as 2018 due to the recency of the NBS term used and have been increasing since. Most of the cases were studied at the catchment scale due to the nature of the modelling tools, and wetlands were by far the most common NBS assessed, followed by bioswales, bioretention cells, and riparian buffers. In terms of modelling, the most common method was using mathematical formulas and custom models to assess the changes in water quality. As for the models themselves, InVEST was employed most often followed by SWMM. It was also found that more than a third of the models employed specifically support the modelling of NBSs or analogue measures. The most commonly studied water quality indicator was Nitrogen in the form of total Nitrogen (TN), which occurred in over half of the publications and was frequently accompanied by total phosphorus (TP) in second place, which appears in half of the papers, with total suspended solids (TSS) coming after that; other elements were less commonly used. Half of the considered publications also performed an economic assessment, most commonly to assess the capital cost of implementation of the solutions, but some also assessed the monetary benefits associated with nutrient pollution reduction using direct benefits, avoided costs, and/or ecosystem service values. Only three papers considered future climate conditions in their studies.

The present study consists of a rapid review, as such, and despite its systematic methodology, it comes with certain limitations. The use of Scopus for the literature search is suitable for the review, but other search engines could have been used to try to obtain a wider range of relevant hits, such as the Web of Science or Google Scholar. As was also referenced in the introduction, the term NBS is still used interchangeably with many others in different areas of study and expanding the search terms to include Sustainable Urban Drainage Systems (SUDS), Water-Sensitive Urban Design (WSUD), and Blue-Green Infrastructure (BGI) was shown to result in a larger selection of papers. The focus of this study is, however, on studies modelling the Impacts of NBS on surface water quality; similar literature reviews have been performed for some of these related nomenclatures as well, as is the case of Eckart et al. [58] who focused on Low-Impact Development (LID) measures.

The main conclusion to be extracted from this study is that NBSs are indeed beneficial to surface water quality, with varying degrees of effectiveness depending on case-specific circumstances and design. The positive impacts of NBS can be observed in the form of reduced nutrient and sediment export as well as the pollution removal effects of specific solutions. The direct and indirect benefits that NBSs can provide to water quality are substantial, and the solutions themselves, when correctly dimensioned and managed, are economically well-justified due to their long-term benefits. Additionally, the studies also suggest that NBS used for water quality improvement will remain an effective strategy in the future, whether under RCP4.5 or RCP 8.5 conditions.

A conclusion that stakeholders can take home from these results is that modelling studies such as the ones reviewed are highly valuable and should be procured prior to the design and implementation of NBS for water quality improvement. Though these solutions are proven to be effective at addressing these issues, each case has its specific context such that no 'one size fits all' solution can be applied. As such, it is important to conduct modelling studies in order to maximize the effectiveness of NBS for each specific case. Additionally, practitioners looking to perform studies on the topic now have more options than ever as varied modelling software suites natively support the modelling of NBS or similar solutions.

As it stands, the knowledge pool on this topic is limited, nonetheless, the available literature does allow for some conclusions to be drawn and serves as a sufficiently effective guide for researchers and practitioners looking to apply modelling methods in their own studies. The literature lacks a solid and diverse pool of studies dealing with the modelling of individual NBS, and so readers may be unable to find concise and replicable methods for studying a specific solution. Additionally, the scales of study observed in the literature tend to be focused either on very large areas or very small ones, meaning there is little knowledge on the topic for several intermediate scales. While there is not necessarily a need for more diversity in case study locations, it would be advisable that more authors that publish water quality modelling studies in America or Asia incorporate the term NBS in their papers alongside the preferred term they may be using to refer to their specific solutions, i.e., this would increase the search results for readers looking for material on this topic. Furthermore, studies modelling the effects of NBS on water quality in future climate conditions are very scarce; this is especially unfortunate as many stakeholders should be looking for this literature as evidence to adopt these measures in an effort to improve resilience. More studies on the subject are necessary to consolidate this knowledge and provide more concrete and replicable methodologies for academics and professionals to use when making such studies.

**Author Contributions:** Conceptualization, methodology, and data curation, F.A.M. and P.R.; formal analysis, investigation, and writing—original draft preparation, F.A.M.; writing—review and editing, supervision, P.R. All authors have read and agreed to the published version of the manuscript.

**Funding:** This research and the APC were funded by the UNaLab project, Grant Agreement No. 730052, Topic: SCC-2-2016-2017: Smart Cities and Communities Nature-based solutions. Thanks

are also due to FCT/MCTES for the financial support to (UIDP/50017/2020 + UIDB/50017/2020), through national funds and co-funding by European funds when applicable.

**Data Availability Statement:** Not applicable.

**Conflicts of Interest:** The authors declare no conflict of interest. The funders had no role in the design of the study; in the collection, analyses, or interpretation of data; in the writing of the manuscript, or in the decision to publish the results.

## Appendix A

**Table A1.** Table showing the titles, authors, and years of publication of the 14 papers used in this literature review.

| Paper Title | Authors | Year |
|---|---|---|
| Ecosystem services evaluation of Nature-Based Solutions with the help of citizen scientists | Di Grazia F., Gumiero B., Galgani L., Troiani E., Ferri M., Loiselle S.A. | 2021 |
| Evaluation of pollutant removal efficiency by small-scale Nature-Based Solutions focusing on bio-retention cells, vegetative swale and porous pavement | Dutta A., Torres A.S., Vojinovic Z. | 2021 |
| Many-Objective Optimization of Sustainable Drainage Systems in Urban Areas with Different Surface Slopes | Seyedashraf O., Bottacin-Busolin A., Harou J.J. | 2021 |
| On the management of Nature-Based Solutions in open-air laboratories: New insights and future perspectives | Gallotti G., Santo M.A., Apostolidou I., Alessandri J., Armigliato A., Basu B., Debele S., Domeneghetti A., Gonzalez-Ollauri A., Kumar P., Mentzafou A., Pilla F., Pulvirenti B., Ruggieri P., Sahani J., Salmivaara A., Basu A.S., Spyrou C., Pinardi N., Toth E., Unguendoli S., Pillai U.P.A., Valentini A., Varlas G., Zaniboni F., Di Sabatino S. | 2021 |
| The impact of bioengineering techniques for riverbank protection on ecosystem services of riparian zones | Symmank L., Natho S., Scholz M., Schröder U., Raupach K., Schulz-Zunkel C. | 2020 |
| Valuing the Multiple Benefits of Blue-Green Infrastructure for a Swedish Case Study: Contrasting the Economic Assessment Tools B£ST and TEEB | Hamann F., Blecken G.-T., Ashley R.M., Viklander M. | 2020 |
| Environmental and Economic Approach to Assess a Horizontal Sub-Surface Flow Wetland in Developing Area | Castañer C.M., Bellver-Domingo Á., Hernández-Sancho F. | 2020 |
| Engaging coastal community members about natural and Nature-Based Solutions to assess their ecosystem function | Baustian M.M., Jung H., Bienn H.C., Barra M., Hemmerling S.A., Wang Y., White E., Meselhe E. | 2020 |
| Ecosystem services from combined natural and engineered water and wastewater treatment systems: Going beyond water quality enhancement | Zawadzka J., Gallagher E., Smith H., Corstanje R. | 2019 |
| Optimizing wetland restoration to improve water quality at a regional scale | Singh N.K., Gourevitch J.D., Wemple B.C., Watson K.B., Rizzo D.M., Polasky S., Ricketts T.H. | 2019 |
| Evaluating the reliability of stormwater treatment systems under various future climate conditions | Zhang K., Manuelpillai D., Raut B., Deletic A., Bach P.M. | 2019 |
| Integrated modelling of stormwater treatment systems uptake | Castonguay A.C., Iftekhar M.S., Urich C., Bach P.M., Deletic A. | 2018 |
| Spatial modelling of the regulating function of the Huangqihai Lake wetland ecosystem | Fu Y., Zhao J., Peng W., Zhu G., Quan Z., Li C. | 2018 |
| Exploring the performances of a new integrated approach of grey, green and blue infrastructures for combined sewer overflows remediation in high-density Urban areas | Masseroni D., Ercolani G., Chiaradia E.A., Maglionico M., Toscano A., Gandolfi C., Bischetti G.B. | 2018 |

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
