# Peer review of "Modelling Impacts of Nature-Based Solutions on Surface Water Quality: A Rapid Review"

_sustainability, doi:10.3390/su14127381_

Round 1

Reviewer 1 Report

The manuscript is dealing with the use of nature based solutions for the water treatment. The topic of the study is contemporary and comes under the purview of the journal. Such kind of approaches are the need of the hour and most of the studies are highlighting it. In this systematic literature survey, authors have find total 18 papers directly linked with the NBS approaches for water treatment/cycling. Indeed the number is too low but the publication of this study may attract wider group of researchers for conducting such kind of studies. As such the manuscript is written well and appropriately presented. However, the focus of the manuscript towards the main content is sometimes seems to be misleading or superficial throughout the different sections. The modeling approach is one of the major attribute of the study, as seems from the title and abstract, but not given much attention in the methodology and other preceding sections. Similarly, conclusion section seems to be repetition of introduction and methodology. It should be more precise and to the point for getting opt information by the readers. The manuscript needs a thorough synthesis for a well-focused review articles. A few minor typological errors and suggestions are highlighted in the attached file for authors' reference.

Author Response

Esteemed reviewer, please find the "Reviewer response" document in attachment.

Reviewer 2 Report

The manuscript documented a rapid systematic review of Scopus-indexed journal articles on the subject of 'Effects of Nature-Based Solutions on Surface Water Quality using Modelling Methods'. The review is of significance and current interest in scientific communication particularly considering approaches to mitigate the effects of current and future climate change scenarios and the need for high-quality evidence to inform implementation, management and policy actions by various stakeholders.

Please consider the comments and edits in the attached file and some are articulated here:

  • The Figures should communicate the intended information without reference to the in-text explanation. Hence, axis labels/titles and keys for acronyms used should be provided.
  • Please edit the manuscript for grammar and spelling. The review results are to be described in reported speech as they were based on studies already conducted and published. Colloquial terms such as 'don't, isn't,...' should be avoided as this is a scientific article, among other edits
  • Please, clearly articulate the research gaps or knowledge to be filled as recommendations for future studies in the conclusion. Also, the take home/uptake for stakeholders should be written clearly, actionable and direct. This is what a systematic review aims to achieve.

Author Response

(The authors gave the same response as above.)

Reviewer 3 Report

The review article is well-organized and I have minor comments before its publication in Sustainability. They are as follows:

- Section 3.2: What we can understand about the Case Study Location? Add some future directions for this part.

- Section 3.6: Add advantages and disadvantages, input data, etc. of  InVest and SWMM models.

Author Response

(The authors gave the same response as above.)

Round 2

Reviewer 1 Report

The manuscript has been revised substantially in light of the suggestions given during previous review. Authors' responses to respective comments/suggestions are satisfactory. I recommend the publication of this manuscript. There are a few typos in the abstract and introduction section which can be curated during the copy-editing and proof-reading stages.

Author Response

We thank the reviewer, once more, for the comments and suggestions provided which helped improve the paper considerably. The paper has since been revised and further improved thanks to the academic editor's notes as well.

Reviewer 2 Report

The manuscript has been adequately revised following the suggested edits and comments. Particularly, the conclusion is been greatly improved highlighting the research gaps and recommendations for future research.

Author Response

(The authors gave the same response as above.)
